# Targeted Therapy of Severe Infections Caused by *Staphylococcus aureus* in Critically Ill Adult Patients: A Multidisciplinary Proposal of Therapeutic Algorithms Based on Real-World Evidence

**DOI:** 10.3390/microorganisms11020394

**Published:** 2023-02-03

**Authors:** Milo Gatti, Bruno Viaggi, Gian Maria Rossolini, Federico Pea, Pierluigi Viale

**Affiliations:** 1Department of Medical and Surgical Sciences, Alma Mater Studiorum University of Bologna, 40138 Bologna, Italy; 2Clinical Pharmacology Unit, Department for Integrated Infectious Risk Management, IRCCS Azienda Ospedaliero-Universitaria di Bologna, 40138 Bologna, Italy; 3Neurointensive Care Unit, Department of Anesthesiology, Careggi, University Hospital, 50134 Florence, Italy; 4Department of Experimental and Clinical Medicine, University of Florence, 50134 Florence, Italy; 5Microbiology and Virology Unit, Florence Careggi University Hospital, 50134 Florence, Italy; 6IRCCS Fondazione Don Carlo Gnocchi, 50143 Florence, Italy; 7Infectious Diseases Unit, Department for Integrated Infectious Risk Management, IRCCS Azienda Ospedaliero-Universitaria di Bologna, 40126 Bologna, Italy

**Keywords:** targeted antibiotic therapy, critically ill patients, MRSA, MSSA, multidisciplinary taskforce

## Abstract

(1) Introduction: To develop evidence-based algorithms for targeted antibiotic therapy of infections caused by *Staphylococcus aureus* in critically ill adult patients. (2) Methods: A multidisciplinary team of four experts had several rounds of assessment for developing algorithms concerning targeted antimicrobial therapy of severe infections caused by *Staphylococcus aureus* in critically ill patients. The literature search was performed by a researcher on PubMed-MEDLINE (until August 2022) to provide evidence for supporting therapeutic choices. Quality and strength of evidence was established according to a hierarchical scale of the study design. Two different algorithms were created, one for methicillin-susceptible *Staphylococcus aureus* (MSSA) and the other for methicillin-resistant *Staphylococcus aureus* (MRSA). The therapeutic options were categorized for each different site of infection and were selected also on the basis of pharmacokinetic/pharmacodynamic features. (3) Results: Cefazolin or oxacillin were the agents proposed for all of the different types of severe MSSA infections. The proposed targeted therapies for severe MRSA infections were different according to the infection site: daptomycin plus fosfomycin or ceftaroline or ceftobiprole for bloodstream infections, infective endocarditis, and/or infections associated with intracardiac/intravascular devices; ceftaroline or ceftobiprole for community-acquired pneumonia; linezolid alone or plus fosfomycin for infection-related ventilator-associated complications or for central nervous system infections; daptomycin alone or plus clindamycin for necrotizing skin and soft tissue infections. (4) Conclusions: We are confident that targeted therapies based on scientific evidence and optimization of the pharmacokinetic/pharmacodynamic features of antibiotic monotherapy or combo therapy may represent valuable strategies for treating MSSA and MRSA infections.

## 1. Introduction

*Staphylococcus aureus* represents one of the major causes of infection among critically ill patients, with a remarkable burden of morbidity and mortality [1,2]. A recent study conducted among 15,202 critically ill patients admitted to 1150 different intensive care units (ICUs) found that 37.0% of clinical isolates retrieved from cultures were Gram-positives, of which *S. aureus* represented more than 50% [3].

Resistance rates of *S. aureus* in Europe is progressively increasing, and currently the proportion of isolates resistant to methicillin (MRSA) is higher than 15% [4]. Additionally, a worrisome finding of the recent pandemic was the remarkable prevalence of documented *S. aureus* superinfections, mainly bloodstream infections (BSIs) and ventilator-associated pneumonia (VAP), that occurred among critically ill patients with COVID-19 [5,6].

ICU patients with *S. aureus* disease can be broadly split into two major categories: on the one hand, those admitted to the ICU because of primary invasive *S. aureus* infection (namely bacteremia, pneumonia, toxic shock syndrome, and severe necrotizing skin and soft-tissue infections), and on the other hand, those with the onset of hospital-associated *S. aureus* infections during the ICU stay, as it may occur typically because of >48 h of ICU stay, recent surgery, and/or implantation of intravascular devices [7]. Sometimes *S. aureus* may produce biofilm, namely a structurally complex extracellular polymeric matrix within which it may embed, and this may render microbiological eradication extremely challenging [2,7]. However, this feature is a common occurrence in chronic infections (i.e., osteomyelitis and orthopedic prosthetic-associated infections), which are rarely encountered in the ICU scenario [2].

Early and prompt anti-MRSA therapy has become a standard approach to face suspected staphylococcal infections in the ICU patients [8]. Dedicated bundles for the treatment of BSIs caused by *S. aureus* were shown to be associated with favorable impact on survival rate [9]. The positive impact of bundles could be furtherly increased if antimicrobial selection were based also on some pharmacokinetic/pharmacodynamic (PK/PD) features, namely the penetration rate and the probability of optimal PK/PD target attainment at the infection site [10]. Rapid diagnostic tests, by providing real-time identification of the pathogen and of resistance markers, could be of further benefit [10,11,12].

In this regard, the establishment of a coordinated multidisciplinary task force, composed of the intensive care physician, the infectious disease consultant, the clinical microbiologist, and the MD clinical pharmacologist, could represent an innovative team focused on maximizing antibiotic efficacy, reducing overconsumption, and minimizing the development of antimicrobial resistance in the ICU setting [9,13].

The aim of this multidisciplinary opinion article was to develop algorithms for targeted antibiotic therapy of infections caused by methicillin susceptible *S. aureus* (MSSA) and MRSA in critically ill patients. The objective was to provide a useful guidance for clinicians in selecting treatment options based on scientific evidence, PK/PD properties of antimicrobials and site of infection.

## 2. Materials and Methods

A multidisciplinary team, composed of one intensive care physician (B.V.), one infectious disease consultant (P.V.), one clinical microbiologist (G.M.R.), and one MD clinical pharmacologist (F.P.), had several virtual meetings for developing algorithms for targeted antimicrobial therapy of severe MSSA and MRSA infections. Algorithms were organized for different sites of infection (namely infective endocarditis [IE], primary bloodstream infections [BSIs], infections associated with intracardiac/intravascular devices, community-acquired pneumonia [CAP], infection-related ventilator-associated complications (IVACs), central nervous system [CNS] infections, and necrotizing skin and soft tissue infections), and therapeutic strategies were based also on PK/PD features [10,14]. A researcher (M.G.) retrieved the scientific evidence for supporting the specific proposals by means of a PubMed-MEDLINE literature search (until August 2022). Key terms concerning selected antibiotics, genotype of resistance and/or antibiotic susceptibility pattern of *S. aureus*, and site of infections were searched in combination. The searched terms were oxacillin, cefazolin, daptomycin, fosfomycin, ceftobiprole, ceftaroline, vancomycin, teicoplanin, linezolid, methicillin-susceptible *Staphylococcus aureus*, methicillin-resistant *Staphylococcus aureus*, MRSA, MSSA, bloodstream infections, bacteremia, infective endocarditis, cardiac implantable electronic device infections, community-acquired pneumonia, hospital-acquired pneumonia, nosocomial pneumonia, pneumonia, ventilator-associated pneumonia, infection-related ventilator-associated complications, central nervous system infections, ventriculitis, meningitis, complicated skin and soft tissue infections, necrotizing soft tissue infections, necrotizing fasciitis, intensive care unit, and critically ill patients. Quality of evidence was established according to a hierarchical scale of the study design, as reported in the evidence pyramid [15]: randomized controlled trials (RCTs), prospective observational studies, retrospective observational studies, case series, case reports, and in vivo/in vitro preclinical studies. International guidelines/guidance documents issued by the Infectious Disease Society of America and/or by the European Society of Clinical Microbiology and Infectious Diseases, systematic reviews, and meta-analyses were also consulted. Consistence between retrieved studies was also considered by assessing the concordance in clinical outcome of the included studies at each level of the evidence pyramid. Only articles published in the English language were considered, with a main focus on those published in the last ten years.

Agreement between all of the four team members was reached on each of the options provided in the therapeutic algorithms after thoroughly discussion based on specific long-standing experience and expertise of each single member.

## 3. Targeted Treatment of Infections Caused by *Staphylococcus aureus* in Critically Ill Patients

Two targeted treatment algorithms were developed, one for MSSA infections and the other for MRSA infections, respectively.

### 3.1. Targeted Treatment of MSSA Infections

Oxacillin (2–3 g q6 h over 6 h by continuous infusion [CI] after 2 g loading dose [LD]) or cefazolin (2 g LD followed by 6–8 g/day by CI) are suggested as targeted therapy for the management of all types of infection caused by MSSA in critically ill patients. The types of infection taken into account were primary or catheter-related BSI, IE, infections associated with intracardiac devices, CAP, IVACs, CNS infections, and necrotizing fasciitis (Figure 1). The scientific evidence supporting these choices was summarized in Table 1.

In regard to BSI, a recent meta-analysis of ten observational studies compared the efficacy of cefazolin and oxacillin in the management of primary and secondary BSIs (34% skin and soft tissue infections, 27% IEs, and 18% pneumonia) caused by MSSA [16]. Cefazolin was significantly associated with lower mortality rate (RR 0.78; 95%CI 0.69–0.88), higher clinical cure rate (RR 1.09; 95%CI 1.02–1.17), and lower risk of drug withdrawal for adverse events (AEs; RR 0.27; 95%CI 0.16–0.47) compared to oxacillin [16]. Conversely, no difference in the relapse rate of BSIs was found between cefazolin and oxacillin treatment (RR 1.29; 95%CI 0.96–1.74). These findings were confirmed in a large multicentric retrospective study that included 3167 patients affected by primary or secondary MSSA BSIs (ICU admission 17.6%) [17]. In that study, cefazolin showed a significantly lower mortality rate both at 30-day (aHR 0.63; 95%CI 0.51–0.78) and at 90-day (aHR 0.77; 95%CI 0.66–0.90) compared to oxacillin or nafcillin. No difference in the relapse rate of BSI was found between the two therapeutic regimens (aOR 1.13; 95%CI 0.94–1.36). Conversely, a recent large multicentric retrospective study including 7312 patients with MSSA BSIs (ICU admission 13.5%) showed no difference in the 30-day mortality rate between cefazolin and flucloxacillin at propensity adjusted analysis (aOR 0.86; 95%CI 0.65–1.14) [18]. Likewise, no difference between cefazolin and antistaphylococcal penicillins (ASPs) was documented in several retrospective observational studies (ICU admission ranging from 5.2% to 41.8%) in terms of mortality rate, clinical cure rate, and relapse or recurrence rate of MSSA BSIs [19,20,21,22,23]. In a multicentric retrospective cohort study including 93 patients with complicated MSSA BSIs (41% related to bone and joint infections and 20% IE) [24], cefazolin showed a significantly lower clinical failure rate at 90-day (24% vs. 47%; *p* = 0.04) and AE rate (3% vs. 30%; *p* < 0.001) compared to oxacillin.

In regard to CNS infections, it should be mentioned that a study carried out among 17 patients with CNS infections caused by MS staphylococcal isolates showed that the percentage of cerebrospinal fluid (CSF) penetration could be higher for cefazolin than for cloxacillin (4.3% vs. 1.8%) [25].

Overall, oxacillin and cefazolin could be considered equivalent for the management of MSSA infections. Several real-world studies showed no significant difference between the two agents in terms of mortality, clinical cure, microbiological eradication, and relapse rate. However, cefazolin could provide some advantages in case of primary and/or secondary BSIs or CNS infections. In regard to the drug administration mode, it is worth mentioning that a retrospective cohort study including 107 patients with MSSA IE found that administration by CI was the only independent variable associated with 30-day microbiological cure with oxacillin at multivariate analysis (*p* = 0.01) and that CI granted significantly higher microbiological cure rate compared to intermittent infusion (94% vs. 79%; *p* = 0.03) [26]. These findings could support the use of CI for maximizing the achievement of optimal PK/PD with oxacillin. Notably, combination therapy did not add any advantage over cefazolin or oxacillin monotherapy in the management of MSSA infections. A recent meta-analysis of 12 studies found that combination therapy did not reduce significantly the mortality rate at day 30 (RR 0.92; 95%CI 0.70–1.20), at day 90 (RR 0.89; 95%CI 0.74–1.06), or at any time (RR 0.91; 95%CI 0.76–1.08) in the treatment of MSSA bacteremia compared to monotherapy [27]. Furthermore, combination therapy was associated with a significantly higher risk of AEs (including nephrotoxicity) (RR 1.74; 95%CI 1.31–2.31; *p* < 0.001). Additionally, a retrospective observational study including 350 patients with MSSA bacteremia (ICU admission 19.5%) found that combination therapy of daptomycin with a beta-lactam failed in reducing mortality rate compared to beta-lactam monotherapy after propensity score-matched analysis (90-day mortality rate: HR 0.89; 95%CI 0.54–1.49) [28].

### 3.2. Targeted Treatment of MRSA Infections

The therapeutic algorithm for targeted treatment of infections caused by MRSA in critically ill patients was organized according to the types of infection site and is shown in Figure 2. The scientific evidence supporting the different choices is summarized in Table 2.

#### 3.2.1. Primary BSIs, Infective Endocarditis, and Intracardiac/Intravascular Devices Infections

Combination therapy with high-dose daptomycin (10 mg/kg q24 h) plus CI fosfomycin (16 g/day after 6–8 g LD) or an anti-MRSA cephalosporin (ceftaroline 600 mg q8 h over 8 h after 600 mg LD or ceftobiprole 500 mg q8 h over 8 h after 500 mg LD) is suggested as first-line targeted therapy of primary BSIs (including catheter related), IEs, or infections associated with intracardiac/intravascular devices caused by MRSA. Vancomycin (2 g/day by CI after 2 g LD) or teicoplanin (6 mg/kg q12 h after an LD of 12 mg/kg q12 h for 4–5 doses) should be reserved as second line alternatives and could be suitable whenever vancomycin MIC is ≤1 mg/L (Figure 2).

In a recent RCT [29], 155 patients with MRSA BSI were randomized to receive daptomycin in monotherapy or in combination therapy with fosfomycin. Combination therapy granted significantly lower rates of both complicated BSI (16.2% vs. 32.1%; *p* = 0.022) and 6-week microbiological failure (0% vs. 11.1%; *p* = 0.003) compared to monotherapy. Additionally, in patients receiving combination therapy a trend toward higher treatment success rate at 6-week was found (54.1% vs. 42.0%; *p* = 0.14). Conversely, combination therapy was associated with significantly higher AE rate (17.6% vs. 4.9%; *p* = 0.018). In the subgroup analysis of patients with Pitt score > 1, a significantly higher cure rate was observed with combination therapy compared to monotherapy (57.1% vs. 22.7%; *p* = 0.023). This is in agreement with the findings of an early case series of three patients with staphylococcal IEs, one of whom had prosthetic valve endocarditis (PVE), who were successfully treated with daptomycin plus fosfomycin [30].

In regard to preclinical models, a rabbit model of experimental MRSA endocarditis investigated the efficacy of daptomycin plus fosfomycin vs. daptomycin alone and showed that combo therapy granted significantly better efficacy in terms of both proportion of sterile vegetations (100% versus 72%; *p* = 0.046) and bacterial burden reduction within the vegetations (*p* = 0.025) [31].

Several clinical studies may support the efficacy of combining daptomycin with ceftaroline in the management of MRSA BSIs, IE, or PVE [32,33,34,35,36,37,38,39,40,41]. An RCT including 40 patients randomized to combination therapy with daptomycin plus ceftaroline or to daptomycin/vancomycin monotherapy of MRSA BSIs (ICU admission up to 18%) showed that combination therapy was associated with significantly lower in-hospital mortality rate (0% vs. 26%; *p* = 0.029) [32]. The surprising findings led the investigators to stop early the study due to an unacceptable higher risk of mortality in the monotherapy arm. Conversely, two large retrospective studies found no difference in terms of mortality rate, hospital readmission, and BSI recurrence of combination therapy with daptomycin plus ceftaroline vs. standard of care monotherapy with vancomycin, daptomycin, or ceftaroline in the treatment of MRSA BSI [33,34]. However, in a subgroup of patients with severe disease (median Charlson Comorbidity Index ≥ 3) and with a primary endovascular source of infection, treatment with daptomycin plus ceftaroline within 72 h from index cultures resulted in a trend toward reduced mortality rate (4.3% vs. 20.8%; *p* = 0.16) [33]. A retrospective cohort study compared combination of daptomycin plus ceftaroline vs. standard of care with daptomycin or vancomycin plus gentamycin or rifampicin in the treatment of complicated MRSA BSI (ICU admission up to 57%) [35]. Combo therapy with daptomycin plus ceftaroline was associated with 77% lower odds of clinical failure at multivariate analysis (OR 0.23; 95%CI 0.06–0.89) and with a significantly lower risk of 60-day recurrence (0% vs. 30%; *p* < 0.01). Cunha et al. reported one case of MRSA aortic PVE successfully treated with a 6-week combination therapy course of high-dose daptomycin (i.e., 12 mg/kg/day) plus ceftaroline [41].

Real-world clinical evidence supporting the use of combination therapy with daptomycin plus ceftobiprole for MRSA BSIs and/or IE is limited only to one case series and one single case report [42,43]. In a case series of 12 staphylococcal IE (of which two-thirds were PVE) treated with ceftobiprole (5/12 underwent surgical intervention because of vegetation size or severe valve dysfunction with heart failure), 11/12 were treated with ceftobiprole plus daptomycin with a clinical cure rate of 83% and none with BSI relapse [42]. Interestingly, an in vitro study assessing the synergic activity of four different antibiotic combinations against 20 MRSA strains found that combo of ceftobiprole plus daptomycin was always synergic against all of the evaluated strains with a four-fold MIC decrease [44].

In regard to glycopeptides, a large multicenter retrospective study assessed 7411 patients with MRSA BSIs treated with vancomycin (n = 6805) or with vancomycin switched to daptomycin (n = 606) during the first hospitalization (108 in the first three days) [45]. MIC for vancomycin was >1 mg/L in 8.2–16.0% of isolates. The overall 30-day mortality rate did not differ between the two groups (aHR 0.87; 95%CI 0.69–1.09) but resulted significantly lower in the subgroup of patients having early switch to daptomycin (aHR 0.48; 95%CI 0.25–0.92). A recent RCT randomized 352 patients with MRSA BSIs to daptomycin/vancomycin monotherapy or daptomycin/vancomycin combined with an anti-staphylococcal beta-lactam (oxacillin, cloxacillin, or cefazolin). No difference between groups was found in the primary composite outcome (90-day mortality rate, relapse, microbiological failure rate, or persistent BSI) (mean difference −4.2%; 95%CI −14.3–6%), but patients receiving combination therapy had significantly higher risk of acute kidney injury (23% vs. 6%; *p* < 0.001) [46].

Overall, combination therapy with daptomycin plus fosfomycin or ceftaroline/ceftobiprole could be the preferred therapeutic strategy for the management of BSIs, IEs and/or infections associated with intracardiac/intravascular devices caused by MRSA. These combinations could grant lower risk of both clinical/microbiological failure and recurrence compared to monotherapy. In regard to PVE, a recent meta-analysis of four studies found no benefits of either gentamicin (OR 0.98; 95%CI 0.39–2.46) or rifampicin (OR 1.29; 95%CI 0.71–2.33) addition in reducing clinical failure in the treatment of staphylococcal PVE [47]. Conversely, the unfavorable PK/PD profile coupled with the safety issues should limit the role of glycopeptides in the treatment critically ill patients with MRSA BSIs, IEs, and/or infections associated with intracardiac/intravascular devices.

#### 3.2.2. Community-Acquired Pneumonia

CI ceftaroline (600 mg q8 h over 8 h after 600 mg LD) or CI ceftobiprole (500 mg q8 h over 8 h after 500 mg LD) are suggested as first-line therapy of severe CAP in patients with risk factors or clinical/radiological features evocative for MRSA etiology (Figure 2).

A recent meta-analysis of 14 studies in patients with pneumonia (of which five were retrospective and provided data on clinical outcome in 345 patients with documented MRSA pneumonia) reported that the pooled success rate of ceftaroline was of 71.7% (95%CI: 59.7–82.3) [48]. A retrospective cohort study including 89 patients affected by severe CAP (10.1% due to MRSA; ICU admission 37%) treated with ceftaroline found a clinical failure rate of 36% [49]. Interestingly, the only independent predictor of clinical failure was the time elapsed from the diagnosis of severe CAP and the start of treatment with ceftaroline (OR for each passing day 1.5; 95%CI 1.1–1.9; *p* = 0.003) [49].

One RCT [50] and one retrospective observational study [51] may support the use of ceftobiprole for severe MRSA CAP. A phase III RCT [50] randomized 638 patients with CAP (4% bacteraemic; 2% MRSA) to ceftobiprole or ceftriaxone plus linezolid. No significant difference between the two groups was found in terms of overall clinical cure rate (86.6% vs. 87.4%) and/or microbiological eradication rate (88.2% vs. 90.8%). A retrospective cohort study [51] including 29 patients treated with ceftobiprole (19.3% CAP; septic shock 13.8%), of whom 24% were affected by MRSA pneumonia, found a clinical cure rate of 68.9% (66.7% in the subgroup of patients affected by CAP). No clinical failure occurred in the subgroup of patients affected by MRSA pneumonia.

Overall, ceftaroline and ceftobiprole could be reasonable options for the treatment of severe MRSA CAP in critically ill patients. Both agents showed similar penetration rate in the epithelial lining fluid (ELF) (20–25%) [52,53]. In regard to mode of administration, some studies suggested that in the treatment of critically ill patients, CI may allow the attainment of higher PK/PD targets with ceftaroline against MRSA compared to intermittent infusion [54,55]. Likewise, the same could be anticipated for ceftobiprole, even if real-world evidence is currently lacking.

#### 3.2.3. Infection-Related Ventilator-Associated Complications

Linezolid (600 mg q12 h) in monotherapy or in combination therapy with CI fosfomycin (16 g q24 h after 6–8 g LD) is suggested as first-line targeted therapy of IVACs caused by MRSA. CI ceftaroline (600 mg q8 h over 8 h after 600 mg LD) or CI ceftobiprole (500 mg q8 h over 8 h after 500 mg LD) could be suggested as potential alternatives, especially in case of bacteraemic infections [56] (Figure 2).

Two meta-analyses [57,58] may support the use of linezolid for treating IVACs due to MRSA. In the first one, Kato et al. [57] analyzed seven RCTs and eight retrospective cohort and/or case-control studies comparing linezolid and vancomycin for the treatment of hospital-acquired or ventilator-associated pneumonia caused by MRSA. Clinical cure and microbiological eradication rates were significantly increased in patients treated with linezolid in both the RCTs (clinical cure: RR 0.81; 95%CI 0.71–0.92; microbiological eradication: RR 0.71; 95%CI 0.62–0.81) and the retrospective studies (clinical cure: OR 0.35; 95%CI 0.18–0.69). Mortality rates were comparable between the two groups in both the RCTs (RR 1.08; 95%CI 0.88–1.32) and the retrospective studies (OR 1.20; 95%CI 0.94–1.53), and no significant difference emerged in terms of AEs. In the second one, Jiang et al. included 12 RCTs comparing linezolid with glycopeptides for the treatment of nosocomial pneumonia [58]. Linezolid was associated with better microbiological eradication rate compared to glycopeptides (RR 1.16; 95%CI 1.03–1.31; *p* = 0.01), whereas no statistically significant difference was found in terms of either clinical cure rate (RR 1.08; 95%CI 1.00–1.17; *p* = 0.06) or all-cause mortality (RR 0.95; 95%CI 0.83–1.09; *p* = 0.46) between the two groups. A higher risk of rash (RR 0.41; 95%CI 0.24–0.71; *p* = 0.001) and nephrotoxicity (RR 0.41; 95%CI 0.27–0.64; *p* < 0.0001) was observed with glycopeptides.

Only preclinical studies may support the role of combining fosfomycin with linezolid for the treatment of MRSA infections [59,60,61,62,63]. A systematic review of seven in vitro preclinical studies documented a synergic effect between linezolid and fosfomycin against 95% of 166 *Staphylococcus aureus* strains, regardless of being planktonic or embedded within the biofilm [59]. In the same review, two in vivo preclinical studies were reported to grant higher efficacy when linezolid was combined with fosfomycin compared to each single agent alone [59].

One meta-analysis [48] and one retrospective cohort study (CAPTURE) [64] may support the use of ceftaroline for the management of severe HAP due to MRSA, including IVACs. A recent meta-analysis of 14 studies in patients with pneumonia (of which five were retrospective and provided data on clinical outcome in 345 patients with documented MRSA pneumonia) reported that the pooled success rate of ceftaroline was 83.0% (95%CI: 65.0–95.0) in HAP/VAP and 71.7% (95%CI: 59.7–82.3) in MRSA pneumonia [48]. A retrospective cohort study including 40 patients affected by HAP/VAP treated with off-label ceftaroline (67.5% ICU admission) found that the clinical success rate was 75% overall (81% and 62% for HAP and VAP, respectively) and 58% in the MRSA subgroup [64].

One post hoc retrospective analysis of a phase III RCT [65], one retrospective cohort study [51], and one in vitro study [66] may support the use of ceftobiprole for severe hospital-acquired pneumonia (HAP) due to MRSA, including IVACs. A post hoc analysis of a phase III RCT including 307 high-risk patients with HAP found that early response (at day 4) was higher among ceftobiprole-treated patients than among comparator-treated patients (12.5% difference; 95%CI 3.5–21.4) [65]. A retrospective cohort study [51] including 29 patients treated with ceftobiprole (48.3% HAP, of which one case of IVAC; septic shock 13.8%), of which 24% were affected by MRSA pneumonia, found a clinical cure rate of 68.9% (85.7% in the subgroup of patients affected by HAP). No clinical failure occurred in the subgroup of patients with MRSA pneumonia. Among 66 MRSA isolates retrieved from HAP, 63/66 (95.5%) were susceptible to ceftobiprole (MIC50 1 mg/L; MIC90 2 mg/L), and 3/66 were ceftobiprole-resistant but with a borderline MIC of 4 mg/L [66].

Overall, linezolid could be considered the preferred therapeutic option for the management of IVACs caused by MRSA. Real-world evidence showed higher clinical cure rate and microbiological eradication rate compared to vancomycin. Linezolid showed very high penetration rate into the ELF of critically ventilated patients (97%), and this may grant effective PK/PD target attainment against MRSA with an MIC up to 2 mg/L [67,68,69]. Fosfomycin showed high penetration rate into the ELF of critically septic patients (>50%) [70] and may theoretically represent a valuable add-on for combo therapy with linezolid according to even physicochemical properties and synergic activity. Novel anti-staphylococcal cephalosporins could be a promising but second-line alternative for IVACs due to MRSA, considering that studies supporting the use of these agents in ICU scenario are limited (mainly assessed in CAP) compared to those available for linezolid. ELF penetration rates were quite limited (20–25%) [52,53], but CI administration may be suggested for maximizing PK/PD target attainment in ventilated critical care patients with normal and/or augmented renal clearance [55].

#### 3.2.4. Central Nervous System Infections

Linezolid (600 mg q12 h) is suggested in monotherapy as targeted therapy of primitive CNS infections caused by MRSA (Figure 2) and in combination therapy with fosfomycin (16 g/day CI after 6–8 g LD) as targeted therapy of post-operative neurosurgical MRSA infections.

Three retrospective studies, two case series, and two case reports may support the use of linezolid for the management of CNS infections caused by MRSA [71,72,73,74,75,76,77,78]. In a retrospective cohort of 66 patients with MRSA CNS infections (78.8% bacteraemic), salvage therapy with linezolid after previous failure of glycopeptide treatment had good efficacy with an in-hospital mortality rate of 13.6% and a relapse rate of 16.7% [71]. Among 26 patients treated with linezolid for *Staphylococcus aureus* meningitis (81% post-operative), the clinical cure rate was 69%, and the microbiological eradication rate was 93% [72]. A retrospective case-control study including 17 patients with MRSA meningitis found significantly higher microbiological cure rate at 5-day among those receiving linezolid compared to those receiving vancomycin (77.8% vs. 25.0%; *p* = 0.044) [73].

Only preclinical studies may support the role of combining fosfomycin with linezolid in treating MRSA infections, as previously mentioned [59,60,61,62,63].

Overall, linezolid could be considered the preferred therapeutic option for the management of CNS infections caused by MRSA. Real-world evidence showed greater clinical cure rate and microbiological eradication rate compared to glycopeptides. Linezolid showed good penetration rate into the cerebrospinal fluid (CSF) of neurocritical patients (57%) [79] and is expected to attain optimal PK/PD targets at the infection site. High-dose fosfomycin (i.e., 24 g/day) may theoretically represent a valuable add-on for combination therapy with linezolid in case of post-operative neurosurgical MRSA infections thanks to a good CSF penetration rate (27%) and to the fact that CSF concentrations were shown to remain above the clinical breakpoint for MRSA during the overall treatment period [80].

#### 3.2.5. Necrotizing Skin and Soft Tissue Infections

High-dose daptomycin in monotherapy (10 mg/kg q24 h) or in combination therapy with clindamycin (600 mg q6 h) is suggested as first-line targeted therapy of necrotizing skin and soft tissue infections caused by MRSA (Figure 2).

A recent meta-analysis of seven studies (907 patients) compared daptomycin and vancomycin in the treatment of bacteraemic MRSA infections and found that treatment success rate with daptomycin was significantly higher in intermediate-risk sources, of which 28–33% were soft tissue infections, (OR 4.40; 95%CI 2.06–9.40; *p* < 0.001) [81]. A large European registry including 6075 patients treated with daptomycin, of which 1927 had complicated skin and soft tissue infections (31.5% MRSA), reported an overall clinical success of 84.7% (87.0% in subgroup of patients with MRSA skin and soft tissue infections) [82]. Notably, daptomycin switch to due to previous clinical failure with other antibiotic therapy was reported in as high as 53.7% of patients [82].

A recent retrospective cohort study including 190 patients affected by necrotizing soft tissue infections of lower limbs found that clindamycin was associated with 2.92 times reduced odds of having an amputation when compared with their counterparts, including in adjusted analysis (*p* = 0.015) [83]. No significant difference in mortality rate between patients receiving clindamycin as part of their initial antibiotic regimen and those treated with comparators was reported (8.3% vs. 11.6%; *p* = 0.45) [83].

Interestingly, a retrospective case-control study [84] including 62 critically ill patients affected by necrotizing soft tissue infections (22.6% caused by *Staphylococcus aureus*) found that an intensive multidisciplinary management (including early surgical debridement followed by daily inspection of surgical wounds, close microbiological surveillance, and targeted antibiotic strategy including high-dose daptomycin for MRSA coupled with clindamycin plus anti-pseudomonal beta-lactam) was more effective than standard management (in which on-demand consultation with emergency surgeon and infectious disease specialist coupled with the use of vancomycin as targeted therapy for MRSA), allowing for earlier control of infection and faster reduction of multiple organ dysfunction (ΔSOFA −5.2 ± 3.5 pts. vs. −2.1 ± 3.0 pts.; *p* = 0.003). Additionally, significant lower 7-day (3.1% vs. 20.0%; *p* = 0.049) and ICU mortality rate (15.6% vs. 40.0%; *p* = 0.032) was found in patients receiving intensive multidisciplinary management.

Overall, high-dose daptomycin could be suggested as targeted therapy of severe necrotizing skin and soft tissue infections caused by MRSA. Real-world data reported better clinical outcome compared to vancomycin in bacteraemic cases. Daptomycin showed very high penetration rate (i.e., 70–100%) in soft tissues [85,86], and this could maximize PK/PD target attainment at the infection site.

## 4. Expert Opinion

*S. aureus* represents a leading cause of infection among critically ill patients [3], and the widespread diffusion of MRSA is especially worrisome [4]. Several diagnostic platforms based on molecular methods are currently available for rapid detection of *S. aureus* and associated *mec* genes responsible for the methicillin-resistant phenotype. These systems work with positive blood cultures or directly with some clinical specimens [87] and can be very useful to rapidly assess, with high sensitivity, the presence of MSSA or MRSA as causes of infection, significantly shortening the time to targeted therapy. However, these systems cannot detect resistances to anti-MRSA cephalosporins and to non-beta-lactam agents, which should be tested by conventional phenotypic susceptibility testing.

In this context, the adoption of criteria that may provide the best therapeutic strategy in each of the different challenging scenarios may be fundamental [10,88].

In regard to MSSA infections, CI oxacillin or CI cefazolin monotherapy may represent the best choice for the management of the different types of infection, as no real-world evidence currently support the role of combination therapy.

In regard to MRSA infections, therapeutic choices should be guided by the infection site. Current evidence suggests that combination of high-dose daptomycin plus fosfomycin or ceftaroline/ceftobiprole may have very high probability of favorable clinical outcome and/or microbiological outcome in the treatment of primary BSIs or IE [29,32] and may represent a promising option for the management of infections associated with intracardiac/intravascular devices, including PVE [89]. Conversely, neither gentamicin nor rifampicin were shown to add significant advantages in this scenario [47]. It should also be noticed that several preclinical studies showed synergic activity between daptomycin and fosfomycin or ceftaroline [59,90,91], but the progressive emergence of fosfomycin-resistant and/or ceftaroline-resistant MRSA strains is quite worrisome [92,93,94,95]. Vancomycin and teicoplanin may be potential alternatives but were not considered as preferred agents due to the unfavorable PK/PD features (low bactericidal effect, risk of nephrotoxicity) [96,97,98]. Linezolid may be the preferred option in the treatment of deep-seated MRSA infections similar to CNS infections and/or IVACs based on the high penetration rates into the CSF and/or the ELF and on the favorable PK/PD features and [67,68,79,99]. Ceftaroline or ceftobiprole could be suggested as potential second-line alternatives in MRSA IVACs, especially in case of bacteraemic infections, although no pivotal trials were conducted in ICU scenario and real-world evidence are limited. High-dose daptomycin could be preferred in the treatment of severe MRSA necrotizing skin and soft tissue infections [100], especially when the pathophysiological alterations commonly retrieved in this type of infection may affect optimal PK/PD target attainment [101].

It should be mentioned that in patients allergic to oxacillin and/or cefazolin, good alternative options for the treatment of MSSA infections may be daptomycin, linezolid, or glycopeptides.

In conclusion, we are confident that these algorithm-based strategies could be helpful in improving clinical outcome in the treatment of challenging scenarios of *S. aureus* infections.

## 5. Conclusions

In an era characterized by the widespread diffusion of MRSA, implementing a multidisciplinary task force for targeting therapy in critically ill patients has become a real need. Our approach is focused on mirroring antibiotic therapy and on optimizing PK/PD target attainment at the infection site. These strategies could be helpful either in improving clinical outcome or in minimizing resistance spread. Availability of rapid molecular diagnostic tests for prompt identification of the causative pathogens will be fundamental for implementing antimicrobial stewardship programs based on the proposed algorithms.

## Figures and Tables

**Figure 1 microorganisms-11-00394-f001:**
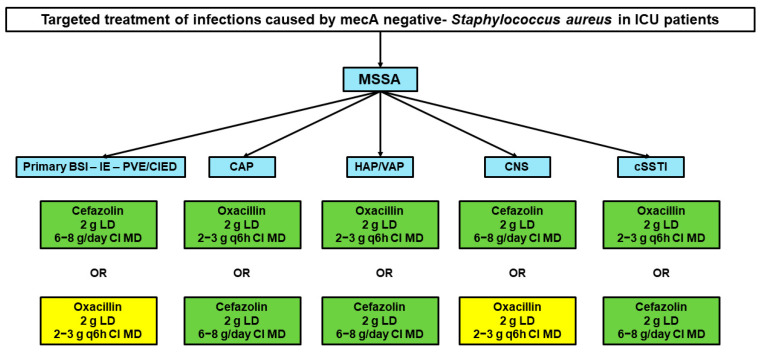
Algorithms for targeted therapy of infections caused by methicillin-susceptible *Staphylococcus aureus*. Green box: best therapeutic regimen according to current evidence; yellow box: alternative therapeutic regimen according to current evidence. BSI: bloodstream infection; CAP: community-acquired pneumonia; CI: continuous infusion; CIED: cardiac implantable electronic device infections; CNS: central nervous system; cSSTI: complicated skin and soft tissue infection; HAP: hospital-acquired pneumonia; ICU: intensive care unit; IE: infective endocarditis; LD: loading dose; MD: maintenance dose; PVE: prosthetic valve endocarditis; VAP: ventilator-associated pneumonia.

**Figure 2 microorganisms-11-00394-f002:**
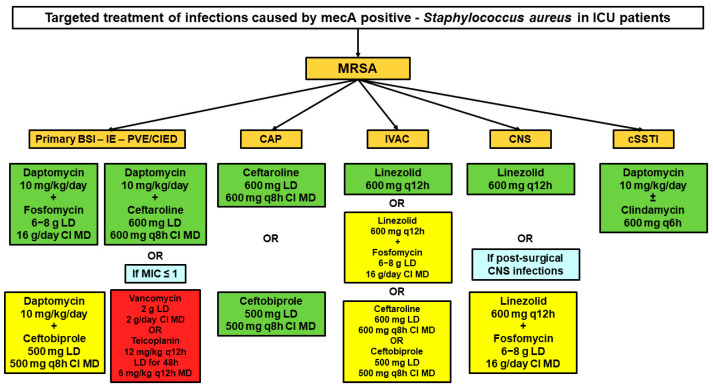
Algorithms for targeted therapy of BSIs or IE caused by methicillin-resistant *Staphylococcus aureus*. Green box: best therapeutic regimen according to current evidence; yellow box: alternative therapeutic regimen according to current evidence; red box: therapeutic regimen recommended only in specific situations. BSI: bloodstream infection; CAP: community-acquired pneumonia; CI: continuous infusion; CIED: cardiac implantable electronic device infections; CNS: central nervous system; cSSTI: complicated skin and soft tissue infection; ICU: intensive care unit; IE: infective endocarditis; IVAC: infective ventilator-associated complications; LD: loading dose; MD: maintenance dose; PVE: prosthetic valve endocarditis.

**Table 1 microorganisms-11-00394-t001:** Summary of studies investigating the treatment of different types of infection caused by methicillin-susceptible *Staphylococcus aureus* with oxacillin or cefazolin.

Author, Year and Reference	Study Design	No. of Patients	Antibiotic and Dosing	Source of Infection	Isolates	Severity	Clinical Outcomes	Relapse Rate—Resistance Development	Comments
Oxacillin or Cefazolin									
Rindone et al., 2018	Systematic review with meta-analysis	10 observational studies (one prospective and nine retrospective)	CEF 3–8 g/day vs. OXA 10–12 g/day	BSIs caused by MSSA Secondary BSI:BJI 9–59% CR-BSI 3–38% SSTI 3–34% LTRI 1–18% IE 0–27%	Significantly lower mortality rate was found with cefazolin vs. oxacillin (RR 0.78; 95%CI 0.69–0.88). Additionally, cefazolin showed significant higher clinical cure rate (RR 1.09; 95%CI 1.02–1.17) and lower risk of withdrawal for AEs (RR 0.27; 95%CI 0.16–0.47). No difference between cefazolin and oxacillin in terms of relapse BSIs was found (RR 1.29; 95%CI 0.96–1.74).				
Davis et al., 2018	Multicentric retrospective cohort	7312 (6520 flucloxacillin vs. 792 CEF)	flucloxacillin vs.CEF	CR-BSI33.2–37.1% SSTI 18.2–18.7% BJI 16.2–18.3% primary 12.4–12.5% IE 5.9–8.0% deep abscess 2.5–2.8% CNS 1.5–2.5%	100% MSSA	ICU admission 13.5–13.9% IHD 8.9–20.8%	30-day mortality rate:11.2% (flucloxacillin) vs. 10.7% (cefazolin)(OR 0.93; 95%CI 0.72–1.17) 30-day mortality rate in propensity adjusted analysis: aOR 0.86 (95%CI 0.65–1.14)	NA	Cefazolin is likely to have equivalent or superior outcomes to ASPs for MSSA bacteraemia.
McDanel et al., 2017	Multicentric retrospective cohort	3167 (1163 CEF vs. 2004 NAF/OXA)	CEF vs. NAF/OXA	SSTI 23–25% BJI 12–13% IE 4–7%	100% MSSA	ICU admission 17.6% APACHE III score > 34 52–56%	30-day mortality rate:aHR 0.63 (95%CI 0.51–0.78) 90-day mortality rate:aHR 0.77 (95%CI 0.66–0.90)	Recurrence: aOR 1.13 (95%CI 0.94–1.36). 90-day MSSA relapse: 2% (CEF) vs. 1% (NAF/OXA)*p* = 0.47 1-year MSSA relapse: 3% (CEF) vs. 2% (NAF/OXA)*p* = 0.07	Patients who received cefazolin had a lower risk of mortality and similar odds of recurrent infections compared with nafcillin or oxacillin for MSSA infections complicated by bacteremia.
Beganovic et al., 2019	Retrospective cohort	212 (105 NAF/OXA vs. 107 CEF)	NAF/OXA vs. CEF	SSTI 21.7% BJI 12.3% surgical site 11.3% IE 7.1% UTI 6.1% LRTI 4.6%	100% MSSA	ICU admission 5.2% Median APACHE score 22–25.5	30-day mortality rate:4.6% vs. 6.8% HR 0.67 (95%CI 0.11–4.00) Discharge: 97.7% vs. 93.2% HR 0.80 (95%CI 0.44–1.44)	30-day readmission: 20.9% vs. 19.5%HR 0.75 (95%CI 0.26–2.16) 30-day reinfection: 9.3% vs. 0.0% *p* = NS	In hospitalized patients with BSIs caused by MSSA, no difference in mortality was observed between NAF/OXA and CEF.
Rao et al., 2015	Multicentric retrospective cohort	161 (103 CEF vs. 58 OXA)	CEF vs. OXA	CR-BSI24.1–45.6% SSTI 14.6–22.4%BJI 13.8–20.4%IE 18.0% LRTI 1.7–1.9% UTI 1.7–1.9% CNS 0.0–1.7%	100% MSSA	ICU admission 32.8–41.8%	Treatment failure rate for deep-seated MSSA infections: 15.6% vs. 20%*p* = 0.72 In-hospital mortality:1% vs. 5.2% *p* = 0.13	BSI recurrence: 4.9% vs. 5.2% *p* = 0.99	Treatment with cefazolin or oxacillin was not independently associated with treatment failure (aOR 3.76; 95%CI, 0.98 to 14.4). Cefazolin was not associated with higher rates of treatment failure and appears to be an effective alternative to oxacillin for treatment of deep-seated MSSA BSI.
Bai et al., 2021	Retrospective cohort	98 (50 CEF vs. 48 cloxacillin)	CEF vs. cloxacillin	Spinal epidural abscesses 100%	100% MSSA	Septic shock 11.2%	90-day mortality rate:8% vs. 13% *p* = 0.52 Failure rate: 12% vs. 19% *p* = 0.21	Recurrence rate: 2% vs. 9% *p* = 0.20 Serious AEs: 0% vs. 4% *p* = 0.24	Cefazolin is likely as effective as an ASP and may be considered as a first-line treatment for MSSA spinal epidural abscesses.
Li et al., 2014	Multicentric retrospective cohort	93 (59 CEF vs. 34 OXA)	CEF 2–8 g/day II or CIvs. OXA 10–12 g/day II or CI	100% complicated BSIBJI 41% IE 20% SSTI 10% CR-BSI 8% UTI 6% LRTI 4% Unknown 11%	100% MSSA	ICU admission 11% Immunosuppression 6%	Clinical cure at EOT:95% vs. 88%*p* = 0.25 30-day mortality rate:0% vs. 3% *p* = 0.37 Overall failure at 90-day: 24% (CEF) vs. 47% (OXA) *p* = 0.04	BSI recurrence:2% vs. 6% *p* = 0.55 AEs rate: 3% vs. 30% *p* < 0.001	Cefazolin appears similar to oxacillin for the treatment of complicated MSSA bacteremia but with significantly improved safety.The higher rates of failure with oxacillin may have been confounded by other patient factors and warrant further investigation.
Corsini Campioli et al., 2021	Retrospective cohort	79 (45 CEF vs. 34 ASPs)	CEF 2 g q8 h vs. OXA 2 g q4 h or NAF 2 g q4 h	Spinal epidural abscesses 100%	100% MSSA	ICU admission 19%	30-day mortality rate:2% vs. 5.9% *p* = 0.57 6-week clinical failure: 75.6% vs. 82.4% *p* = 0.58 12-week clinical failure: 33.3% vs. 44.1% *p* = 0.35	90-day recurrence: 11.4% vs. 9.4%*p* = 0.99	Cefazolin was equally as effective as ASPs, suggesting that it can be an alternative to ASPs in the treatment of MSSA spinal epidural abscesses.
Lefevre et al., 2021	Retrospective cohort	73 (35 ASPs vs. 38 CEF)	ASPs 12 g/day vs. CEF 6 g/day	IE 100%	100% MSSA	Septic shock 30.1%	90-day mortality rate:28.6% (ASPs) vs. 21.1% (CEF) *p* = 0.57	Relapse 0.0% vs. 5.3%*p* = 0.49	Efficacy and safety did not statistically differ between ASPs and cefazolin for MSSA-IE treatment.
Le Turnier et al., 2020	Retrospective cohort	17 (8 CEF vs. 9 cloxacillin)	CEF 2 g q6 h CI vs. cloxacillin 2 g q4 h CI	CNS 100%	58.8% MSSA 35.3% MSSE 5.9% *S. lugdunensis*	NA	Ratio concentration CSF/plasma: 4.3% (CEF) vs. 1.8% (cloxacillin)	Clinical failure:0% (CEF) vs. 22.2% (cloxacillin)	Patients with staphylococcal meningitis treated with high-dose continuous intravenous infusion of CEF achieved therapeutic concentrations in CSF. CEF appears to be a therapeutic candidate which should be properly evaluated in this indication.
Hughes et al., 2009	Retrospective cohort	107	CI OXA (78 patients) vs.II OXA (29 patients)	IE 100%	100% MSSA	IHD 7%	30-day mortality rate:8% (CI) vs. 10% (II) *p* = 0.7 30-day microbiological cure: 94% (CI) vs. 79% (II)*p* = 0.03	NA	CI emerged as the only independent variable associated with 30-day microbiological cure at multivariate analysis (*p* = 0.01). CI oxacillin is an effective alternative to II oxacillin for the treatment of IE caused by MSSA and may improve microbiological cure. This convenient and pharmacodynamically optimized dosing regimen for oxacillin deserves consideration for patients with IE caused by MSSA.

AE: adverse event; ASP: anti-staphylococcal penicillins; BJI: bone and joint infection; BSI: bloodstream infection; CEF: cefazolin; CI: continuous infusion; CNS: central nervous system; CR-BSI: catheter-related bloodstream infection; CSF: cerebrospinal fluid; EOT: end of treatment; HR: hazard ratio; ICU: intensive care unit; IE: infective endocarditis; IHD: intermittent hemodialysis; II: intermittent infusion; LRTI: lower respiratory tract infection; MSSA: methicillin-susceptible *S. aureus*; MSSE: methicillin-susceptible *S. epidermidis*; NAF: nafcillin; NA: not assessed; NS: not significant; OR: odds ratio; OXA: oxacillin; RR: risk ratio; SSTI: skin and soft tissue infection; UTI: urinary tract infection.

**Table 2 microorganisms-11-00394-t002:** Summary of studies investigating the treatment of infection caused by methicillin-resistant *Staphylococcus aureus* according to the different sites of infection.

Author, Year and Reference	Study Design	No. of Patients	Antibiotic and Dosing	Source of Infection	Isolates	Severity	Clinical Outcomes	Relapse Rate— Resistance Development	Comments
Primary or CR-BSI; IE; Infections Associated with Intracardiac Device—Daptomycin + Fosfomycin									
Pujol et al., 2021	RCT	155 (74 DAP + FOS vs. 81 DAP)	DAP 10 mg/kg/day + FOS 2 g q6 h vs. DAP 10 mg/kg/day	100% BSIs CR-BSI 41.9–48.1% SSTI 13.5–23.5% IE 11.1–12.2% surgical site 4.9–9.5% UTI 3.7–8.1% other 7.4–9.9% unknown 9.9–18.9%	100% MRSA	Median CCI: 3–4 Mean Pitt score: 1.15–1.22	Treatment success at 6-week: 54.1% vs. 42.0% (RR 1.29; 95%CI 0.93–1.80; *p* = 0.14) Complicated BSI: 16.2% vs. 32.1% *p* = 0.022	6-week microbiological failure rate: 0% vs. 11.1% *p* = 0.003 AEs rate: 17.6% vs. 4.9%*p* = 0.018	Daptomycin plus fosfomycin provided a 12% higher rate of treatment success than daptomycin alone. This antibiotic combination prevented microbiological failure and complicated BSI, but it was more often associated with AEs.
Mirò et al., 2012	Case series + In vitro study	3 (+14 in vitro tested isolates)	DAP 10 mg/kg/day + FOS 2 g q6 h	100% IEs	67% MRSA 33% MSSA	NA	Clinical cure: 100%	AEs rate: 0.0%	This combination was tested in vitro against 7 MSSA, 5 MRSA, and 2 intermediately glycopeptide-resistant *S. aureus* isolates and proved to be synergistic against 11 (79%) strains and bactericidal against 8 (57%) strains.
Garcia-de-la-Maria et al., 2018	In vivo study (rabbit model)	5 MRSA strains	DAP 6–10 mg/kg + FOS 2 g q6 h or cloxacillin 2 g q4 h	Daptomycin plus fosfomycin significantly improved the efficacy of daptomycin monotherapy at 6 mg/kg/day in terms of both the proportion of sterile vegetations (100% versus 72%, *p* = 0.046) and the decrease in the density of bacteria within the vegetations (*p* = 0.025). Daptomycin plus fosfomycin was as effective as daptomycin monotherapy at 10 mg/kg/day (100% versus 93%, *p* = 1.00) and had activity similar to that of daptomycin plus cloxacillin when daptomycin was administered at 6 mg/kg/day (100% versus 88%, *p* = 0.48). Daptomycin nonsusceptibility was not detected in any of the isolates recovered from vegetations. In conclusion, for the treatment of MRSA experimental endocarditis, the combination of daptomycin plus fosfomycin showed synergistic and bactericidal activity.					
**Primary or CR-BSI; IE; Infections Associated with Intracardiac Device—Daptomycin + Ceftaroline or Ceftobiprole**									
Geriak et al., 2019	RCT	40 (17 DAP + ceftaroline vs. 23 DAP/VAN monotherapy)	DAP 6–8 mg/kg/day + ceftaroline 600 mg q8 h	100% BSIs SSTIs 35–53% BJI 17–29% LRTI 6–26% UTI 17–18% IE 4–19% CR-BSI 6–13% IAI 9%	100% MRSA	ICU admission 13–18%Median CCI 5–6 Immunosuppressed 4%	In-hospital mortality rate: 0% vs. 26% *p* = 0.029	NA	This exploratory study showed with a very small number of patients that initial therapy with DAP + ceftaroline may be associated with reduced in-hospital mortality compared with the treatment standards of VAN or DAP monotherapy in patients with MRSA bacteremia. The survival benefit, if any, may be limited to patients with high-risk endovascular sources and those with IL-10 of >5 pg/mL on the day of first positive blood culture.
McCreary et al., 2019	Retrospective matched cohort study	171 (58 DAP + ceftaroline vs. 113 standard of care)	DAP 8 mg/kg/day + ceftaroline 600 mg q8 h	100% BSIs Endovascular 53% Secondary 42% CR-BSI 5%	100% MRSA	ICU admission 16% CCI >3 49–57% Immunosuppressed 9%	30-day mortality rate: 8.3% vs. 14.2% *p* = NS Lower 30-day mortality rate in subgroup of patients receiving combination therapy with a CCI ≥3, endovascular source, and receipt of DAP-ceftaroline within 72 h of index culture	Relapse: 8.6% vs. 9.7% *p*= NS	DAP-ceftaroline treatment is often delayed in MRSA BSI. Combination therapy may be more beneficial if initiated earlier, particularly in patients at higher risk for mortality.
Nichols et al., 2021	Retrospective case-control study	140 (66 DAP + ceftaroline vs. 74 DAP/VAN or ceftaroline monotherapy)	DAP + ceftaroline	100% BSIs	100% MRSA	ICU admission: 57–64% MV 11–18% Vasopressors: 15–17%Median CCI: 2–3 Immunosuppressed 13%	Primary outcome (infection-related mortality, 60-day readmission, 60-day BSI recurrence): 21% vs. 24% *p* = 0.66	BSI recurrence: 3% vs. 7% *p* = 0.45	No difference was found in the composite outcome of 60-day bacteremia recurrence, readmission, or inpatient infection-related mortality for patients with MRSA bacteremia retained on combination therapy versus those de-escalated to monotherapy.
Johnson et al., 2021	Retrospective cohort study	60 (30 DAP + ceftaroline vs. 30 DAP/VAN ± GEN/RIF)	DAP 10 mg/kg/day + ceftaroline 600 mg q8 h	100% BSIs Endovascular 37–40% IE 23–50% CR-BSI 6.7–37% SSTI 13–23% LRTI 13% IAI 3.3% UTI 3.3%	100% MRSA	ICU admission 53–57% Immunocompromised 10–13% Median CCI 5	Clinical failure rate: 20% vs. 43% *p* = 0.052 At multivariate analysis, DAP + ceftaroline was associated with 77% lower odds of clinical failure (OR 0.23; 95%CI 0.06–0.89)	60-day recurrence: 0% vs. 30% *p* < 0.01	In patients with complicated MRSA-BSI with delayed clearance, DAP + ceftaroline trended towards lower rates of clinical failure than SoC and was significantly associated with decreased clinical failure after adjustment for baseline differences.
Ahmad et al., 2020	Retrospective case-control study	30 (15 DAP/VAN + ceftaroline vs. 15 switched to DAP/VAN monotherapy following BSI resolution)	VAN 15–20 mg/kg q8–12 h or DAP 8–10 mg/kg/day + ceftaroline 600 mg q8–12 h	100% BSIs IE 33–87% BJI 7–47% CNS infection 7%	100% MRSA	Median CCI 0	Mortality rate: 20% vs. 7% *p* = 0.24	Recurrence: 0% vs. 27% *p* = 0.27	In subjects with complicated and prolonged MRSA bacteremia requiring supplemental ceftaroline, clinical outcomes did not differ among patients prescribed DAP/VAN alone following bacteremia resolution vs. patients who continued combination therapy.
Sakoulas et al., 2014	Retrospective multicenter study + *in vitro* analysis	26	DAP 4–10 mg/kg/day + ceftaroline 200 mg q12 h– 600 mg q8 h	100% BSIs 54% IEs 42% BJI 4% SSTI	76.9% MRSA 7.7% MSSA 7.7% VISA 7.7% MRSE	NA	Mortality rate: 4% Time to bacteremia clearance: 10 (previous therapeutic regimens) vs. 2 days (DAP + ceftaroline)		Ceftaroline plus daptomycin may be an option to hasten clearance of refractory staphylococcal bacteremia. Ceftaroline offers dual benefit via synergy with both daptomycin and sensitization to innate host defense peptide cathelicidin LL37, which could attenuate virulence of the pathogen.
Cortes-Penfield et al., 2019	Retrospective cohort study	17 (5 DAP monotherapy vs. 12 DAP + ceftaroline 2–3° line)	DAP 8 mg/kg/day + ceftaroline	100% persistent BSIs BJI 47.1% IE 29.4% SSTI 23.5%	100% MRSA	ICU admission 64.7% Mean CCI: 3.2–5	Mortality rate: 53% Duration of BSI: 11.1 (early combination therapy) vs. 17.3 days *p* = 0.11	NA	Early combination therapy with daptomycin and ceftaroline shortens prolonged MRSA bacteremia and may be helpful in securing favorable clinical outcomes.
Hornak et al., 2019	Case series	11 (6 DAP + ceftaroline; 5 VAN + ceftaroline)	DAP or VAN + ceftaroline 200 mg q12 h– 600 mg q8 h	100% BSIs	100% MRSA	Median CCI 4.5 Immunosuppressed 20%	Microbiological cure rate:100% 30-day mortality rate: 11.1%	30-/60-day relapse: 0.0%	Combination therapy demonstrated success in diverse cases of refractory MRSA BSIs, including instances of persistent bacteremia paired with incomplete source control.
Duss et al., 2019	Case report + in vitro analysis	1	DAP 10 mg/kg/day + ceftaroline600 mg q8 h	IE	MRSA	NA	Clinical cure In in vitro analysis, at high inoculum only combination between DAP and ceftaroline provides synergistic and bactericidal activity	No relapse	A synergistic effect between daptomycin plus ceftaroline and increased bactericidal activity against MRSA was reported, suggesting that this combination may be effective for the treatment of invasive MRSA infection.
Cunha et al., 2015	Case report	1	DAP 12 mg/kg/day + Ceftaroline 600 mg q12 h	PVE	100% MRSA	NA	Clinical cure	No relapse	Ceftaroline plus high-dose daptomycin could be a treatment option for PVE sustained by difficult-to- treat MRSA strains.
Tascini et al., 2020	Case series	12	Ceftobiprole + DAP (in 11 patients)	100% IEs 67% PVEs	33.3% MRSA 33.3% MSSA 33.3% CoNS	Immunosuppressed 16.7%	Clinical cure rate 83%	Relapse 0.0%	Ceftobiprole, especially in combination, could be a promising alternative treatment for infective endocarditis.
Oltolini et al., 2016	Case report	1	DAP 10 mg/kg/day + ceftobiprole 500 mg q8 h	IE	MRSA	NA	Clinical cure	No relapse	Ceftobiprole plus daptomycin could be a treatment option for IE sustained by difficult-to- treat MRSA strains.
Barber et al., 2014	In vitro study	20 MRSA isolates	DAP + ceftobiprole	Ceftobiprole plus daptomycin represented the most potent combination with a 4-fold decrease in MIC and synergy against all strains evaluated in time–kill evaluations. Additionally, binding studies demonstrated enhanced daptomycin binding in the presence of subinhibitory concentrations of ceftobiprole. The use of combination therapy with ceftobiprole may provide a needed addition for the treatment of Gram-positive infections resistant to daptomycin or vancomycin.					
**Primary or CR-BSI; IE; Infections Associated with Intracardiac Device—Vancomycin or Teicoplanin**									
Schweizer et al., 2021	Multicenter retrospective cohort	7411 of which 606 switched to DAP during the first hospitalization and 108 within the first 3 days	VAN vs. switch to DAP	100% BSIs SSTI 46.4–48.5%BJI 20.4–29.2% Endovascular 18.1–29.9% LRTI 2.3–5.0%	100% MRSA MIC > 1 mg/L for VAN: 8.2–16.0%	ICU admission 5.5–7.1% Immunosuppressed 76.2%	30-day mortality rate: 8.3% (early switch to DAP) vs. 17.4% (VAN) aHR 0.48 (95%CI 0.25–0.92) 30-day mortality rate: 12.9% (any switch) vs. 17.4% (VAN) aHR 0.87 (95%CI 0.69–1.09)	NA	Switching to daptomycin within 3 days of initial receipt of vancomycin is associated with lower 30-day mortality among patients with MRSA BSI.
Tong et al., 2020	RCT	352 (174 DAP/VAN + beta-lactam vs. 178 DAP/VAN)	DAP 6–10 mg/kg/day or VAN 1 g q12 h + Oxacillin 2 g q6 h or Cloxacillin 2 g q6 h or CEF 2 g q8 h vs. DAP 6–10 mg/kg/day or VAN 1 g q12 h	100% BSI SSTI 23–28% primary BSI 20% BJI 15–18% CR-BSI 12–14% LRTI 6–7% IE 3–5% other 7–10%	100% MRSA	Median CCI: 5 Median SOFA score: 1–2	Primary outcome (90-day mortality, relapse, persistent BSI, microbiological failure): 35% vs. 39% (−4.2%; 95%CI −14.3% to 6%) 90-day mortality: 21% vs. 16% (4.5%; 95%CI −3.7% to 12.7%)	Persistent BSI: 11% vs. 20% (−8.9%; 95%CI −16.6% to −1.2%) AKI: 23% vs. 6% (17.2%; 95%CI 9.3% vs. 25.2%)	Among patients with MRSA bacteremia, addition of an antistaphylococcal β-lactam to standard antibiotic therapy with vancomycin or daptomycin did not result in significant improvement in the primary composite end point of mortality, persistent bacteremia, relapse, or treatment failure. Early trial termination for safety concerns and the possibility that the study was underpowered to detect clinically important differences in favor of the intervention should be considered when interpreting the findings.
**Community-Acquired Pneumonia—Ceftaroline or Ceftobiprole**									
Sotgiu et al., 2018	Systematic review with meta-analysis	6 retrospective observational studies providing data on patients with documented MRSA pneumonia (345 patients)	Ceftaroline 600 mg q12 h	CAP/HAP/VAP caused by MRSA	Pooled success rate in CAP subgroup: 81.3% (95%CI 80.0–82.7) Pooled success rate in MRSA subgroup: 71.7% (95%CI 59.7–82.3)				
Bassetti et al., 2020	Retrospective cohort study	89	Ceftaroline 600 mg q8 h (60% combination therapy)	100% severe CAP 12% bacteraemic	Isolated pathogens in 34.8% of included cases 10.1% MRSA	ICU admission 37% Septic shock 12% Immunosuppressed 40% Mean CCI 4 ± 3	30-day mortality rate: 20% Clinical failure rate: 36% The only independent predictor of clinical failure was the time elapsing from severe CAP diagnosis to ceftaroline therapy (OR for each passing day 1.5, 95%CI 1.1–1.9, *p* = 0.003).	NA	Ceftaroline could represent an important therapeutic option for severe CAP.
Nicholson et al., 2012	RCT	638 (314 ceftobiprole vs. 324 ceftriaxone plus LIN)	Ceftobiprole 500 mg q8 h vs. Ceftriaxone 2 g/day ± Linezolid 600 mg q12 h	100% CAP 4% Bacteraemic	Isolated pathogens in 28.8% of included cases	PSI ≥ 4: 22% SIRS 52–55%	Clinical cure: 86.6% vs. 87.4% (95%CI −6.9% to 5.3%) Microbiological eradication: 88.2% vs. 90.8% (95%CI −12.6% to 7.5%)	NA	Ceftobiprole was non-inferior to the comparator (ceftriaxone ± linezolid) in all clinical and microbiological analyses conducted, suggesting that ceftobiprole has a potential role in treating hospitalized patients with CAP.
Durante-Mangoni et al., 2020	Retrospective cohort study	29	Ceftobiprole 250 mg/die– 500 mg q8 h	19.3% CAP	24.1% MRSA	Septic shock 13.8%	Clinical cure rate: 68.9% (66.7% in CAP subgroup)	NA	Ceftobiprole, even outside current indications, may be a safe and effective treatment for resistant Gram-positive cocci infections where other drugs are inactive or poorly tolerated and for salvage therapy.
**Infection-Related Ventilator-Associated Complications—Linezolid or Linezolid + Fosfomycin**									
Kato et al., 2021	Systematic review with meta-analysis	7 RCTs (1239 patients) and 8 retrospective observational studies (6125 patients)	LIN 600 mg q12 h vs. VAN 1 g q12 h or 15 mg/kg q12 h	HAP/VAP caused by MRSA	Clinical cure and microbiological eradication rates were significantly increased in patients treated with LIN in RCTs (clinical cure: RR 0.81; 95%CI 0.71–0.92; microbiological eradication: RR 0.71; 95%CI 0.62–0.81) and retrospective studies (clinical cure: OR 0.35; 95%CI 0.18–0.69). However, mortality was comparable between patients treated with VAN and LIN in RCTs (RR 1.08; 95%CI 0.88–1.32) and retrospective studies (OR 1.20; 95%CI 0.94–1.53). Likewise, there was no significant difference in AEs between VAN and LIN in retrospective studies (thrombocytopenia: OR 0.95; 95%CI 0.50–1.82; nephrotoxicity: OR 1.72; 95%CI 0.85–3.45). According to our meta-analysis of RCTs and retrospective studies conducted worldwide, we found robust evidence to corroborate the IDSA guidelines for the treatment of proven MRSA pneumonia.				
Jiang et al., 2013	Systematic review with meta-analysis	12 RCTs (4725 patients)	LIN vs. VAN or TEI	HAP	There was no statistically significant difference between the two groups in the treatment of nosocomial pneumonia regarding the clinical cure rate (RR 1.08; 95%CI 1.00–1.17; *p* = 0.06). Linezolid was associated with better microbiological eradication rate in nosocomial pneumonia patients compared with glycopeptide antibiotics (RR 1.16; 95%CI 1.03–1.31; *p* = 0.01). There were no differences in the all-cause mortality (RR 0.95; 95%CI 0.83–1.09; *p* = 0.46) between the two groups. However, the risks of rash (RR 0.41; 95%CI 0.24–0.71; *p* = 0.001) and renal dysfunction (RR 0.41; 95%CI = 0.27–0.64; *p* < 0.0001) were higher with glycopeptide antibiotics.				
Antonello et al., 2020	Systematic review of in vitro studies	9 in vitro/in vivo preclinical studies	LIN + FOS	*S. aureus* isolates (166 strains)	Combination therapy including FOS and LIN had a synergistic effect in vitro approximately in 95% of cases (synergistic effect of the combination against 100% of the tested isolates was reported in 6 in vitro studies) and even against staphylococcal biofilm cultures. Furthermore, the only 2 in vivo studies performed proved FOS + LZD combination to have higher efficacy than FOS or LIN alone.				
Chen et al., 2018	In vitro study	11 *S. aureus* strains (5 MSSA and 6 MRSA)	LIN + FOS	Synergistic effects were observed for eight strains, and no antagonism was found with any combination. Moreover, LIN combined with FOS at 4× MIC showed the best synergistic antibacterial effect, and this effect was retained after 24 h. In addition, both the antibiotics alone and in combination showed increased post-antibiotic effect and post-antibiotic subminimum inhibitory concentration effect values in a concentration- and time-dependent manner.					
Li et al., 2020	In vitro/in vivo preclinical study	4 *S. aureus* strains (2 MSSA and 2 MRSA)	LIN 10 mg/kg + FOS 200 mg/kg	The combination of linezolid and fosfomycin was synergistic and bacteriostatic against four tested strains. Treatment of *Galleria mellonella* larvae infected with lethal doses of *S. aureus* resulted in significantly enhanced survival rates when low-dose of combination has no significant differences with high-dose combination. Combination therapy including linezolid and fosfomycin has synergistic effect against *S. aureus* in vitro and in an experimental *G. mellonella* model, and it suggests that a high dose of linezolid and fosfomycin may not be necessary.					
Chai et al., 2016	In vitro/in vivo preclinical study	3 MRSA strains	LIN 40 mg/kg q12 h + FOS 300 mg/kg q12 h	A FICI ≤ 0.5 was found for LIN + FOS combination, showing the best synergistic effect in all strains. The combination of LIN and FOS in a catheter-related biofilm rat model found that viable bacteria counts in biofilm were significantly reduced after treatment (*p* < 0.05).					
Xie et al., 2021	In vitro/in vivo preclinical study	One MRSA strain	LIN 2.5–10 mg/kg + FOS 50–200 mg/kg	Antibiotic combination showed excellent synergistic or additive effects on the original and the linezolid-resistant strain but showed indifferent effect for fosfomycin-resistant strain. In the *Galleria mellonella* infection model, the survival rate of the antibiotic combined was improved compared with that of the single drug. There was a good correlation between in vivo efficacy and in vitro susceptibility.					
**Infection-Related Ventilator-Associated Complications—Ceftaroline or Ceftobiprole**									
Sotgiu et al., 2018	Systematic review with meta-analysis	6 retrospective observational studies providing data on patients with documented MRSA pneumonia (345 patients)	Ceftaroline 600 mg q12 h	CAP/HAP/VAP caused by MRSA	Pooled success rate in HAP/VAP subgroup: 83.0% (95%CI 65.0–95.0) Pooled success rate in MRSA subgroup: 71.7% (95%CI 59.7–82.3)				
Kaye et al., 2015	Retrospective cohort study	40	Ceftaroline	67.5% HAP 32.5% VAP	47.5% MRSA	ICU admission 42.5%	Overall clinical cure rate: 75.0% (61.5% in VAP subgroup) Clinical success rate in MRSA subgroup: 57.9%	NA	Ceftaroline is an effective treatment option for HAP and VAP when a susceptible etiologic pathogen is identified, including MRSA.
Scheeren et al., 2019	Retrospective analysis of an RCT	307 (169 ceftobiprole vs.138 comparator)	Ceftobiprole 500 mg q8 h vs. Ceftazidime + Linezolid	100% HAP	23.7% *S. aureus*	Mechanical ventilation 22.5%	Early clinical response (at day 4): difference 12.5% (95%CI 3.5–21.4)	NA	Ceftobiprole treatment may have advantages over other antibiotics in terms of achieving early improvement in high-risk patients with HAP.
Durante-Mangoni et al., 2020	Retrospective cohort study	29	Ceftobiprole 250 mg/die– 500 mg q8 h	47.8% HAP/VAP	24.1% MRSA	Septic shock 13.8%	Clinical cure rate: 68.9% (85.7% in HAP/VAP subgroup) No clinical failure in MRSA subgroup	NA	Ceftobiprole, even outside current indications, may be a safe and effective treatment for resistant Gram-positive cocci infections where other drugs are inactive or poorly tolerated and for salvage therapy.
Antonelli et al., 2019	In vitro study	66 MRSA isolates from HAP	Ceftobiprole	Overall susceptibility to ceftobiprole: 95.5%; MIC_50_: 1 mg/L; MIC_90_: 2 mg/L					
**Central Nervous System Infections—Linezolid or Linezolid + Fosfomycin**									
Chen et al., 2020	Retrospective cohort study	66	LIN	Brain abscess 28.8% Spinal epidural abscess 27.3% Meningitis 18.2% Meningitis + brain/epidural abscess 13.6% Spine device-related infection 7.6%	100% MRSA Bacteraemic 78.8%	Liver cirrhosis 21.2%	In-hospital mortality rate: 13.6%	Relapse rate: 16.7%	LIN demonstrated promising effect as a salvage therapy for central nervous system infection caused by MRSA, whether due to drug allergy or glycopeptide treatment failure.
Pintado et al., 2020	Retrospective multicentric cohort study	26	LIN 600 mg q12 h (62% monotherapy)	100% meningitis 81% post-operative	15 MRSA 11 MSSA Bacteraemic 8%	Immunosuppressed 8%	Clinical cure rate: 69% Microbiological cure rate: 93% 30-day mortality rate: 23%	NA	Linezolid appears to be effective and safe for therapy of *S. aureus* meningitis.
Sipahi et al., 2013	Retrospective case-control study	17	LIN 600 mg q12 h vs. VAN 500 mg q6 h	Meningitis 100%	17 MRSA	NA	Microbiological cure at 5-day: 77.8% vs. 25.0% *p* = 0.044	NA	Findings suggested that LIN is superior to VAN for treating MRSA meningitis, especially in cases in which there is a high MIC (2 mg/L) for VAN.
Rebai et al., 2019	Case series	10	LIN 600 mg q12 h	Meningitis 60%Ventriculitis 20% Subdural empyema 20%	7 MRSA 3 MRSE	NA	Microbiological cure rate: 100%	NA	LIN could be an alternative to VAN for the treatment of post-neurosurgical infections caused by MRSA with a high rate of efficacy.
Viaggi et al., 2011	PK study	7	LIN 600 mg q12 h	100% external ventricular drainage	Prophylaxis of CNS infection	ICU admission 100%	AUC_CSF_/AUC_pkasma_ 0.57	NA	The wide variability in the CSF concentration profile and drug PK among patients suggests the adoption of TDM-guided strategy.
Saito et al., 2010	Case series	2	LIN 600 mg q12 h	100% intracranial abscess	2 MRSA	ICU admission 100%	Clinical cure 100%	None	LIN showed high CSF penetration allowing for the effective treatment of post-neurosurgical infections caused by MRSA.
Kallweit et al., 2007	Case report	1	LIN 600 mg q12 h	Meningitis	MRSA	NA	Clinical cure	None	LIN showed high CSF penetration allowing for the effective treatment of post-neurosurgical infections caused by MRSA.
Kessler et al., 2007	Case report	1	LIN 600 mg q12 h	Meningitis	MRSA	NA	Clinical cure	None	LIN showed high CSF penetration allowing for the effective treatment of post-neurosurgical infections caused by MRSA.
Pfausler et al., 2004	PK study	6	FOS 8 g q8 h	100% ventriculitis	2 MSSA 2 MSSE 2 NA	ICU admission 100%	AUC_CSF_/AUC_pkasma_ 0.27 ± 0.08	NA	High-dose FOS could provide sufficient antimicrobial concentrations in the CSF for the overall treatment period.
Antonello et al., 2020	Systematic review of in vitro studies	9 in vitro/in vivo preclinical studies	LIN + FOS	*S. aureus* isolates (166 strains)	Combination therapy including FOS and LIN had a synergistic effect in vitro approximately in 95% of cases (synergistic effect of the combination against 100% of the tested isolates was reported in 6 in vitro studies) and even against staphylococcal biofilm cultures. Furthermore, the only 2 in vivo studies performed proved FOS + LZD combination to have higher efficacy than FOS or LIN alone.				
Chen et al., 2018	In vitro study	11 *S. aureus* strains (5 MSSA and 6 MRSA)	LIN + FOS	Synergistic effects were observed for eight strains, and no antagonism was found with any combination. Moreover, LIN combined with FOS at 4× MIC showed the best synergistic antibacterial effect, and this effect was retained after 24 h. In addition, both the antibiotics alone and in combination showed increased post-antibiotic effect and post-antibiotic subminimum inhibitory concentration effect values in a concentration- and time-dependent manner.					
Li et al., 2020	In vitro/in vivo preclinical study	4 *S. aureus* strains (2 MSSA and 2 MRSA)	LIN 10 mg/kg + FOS 200 mg/kg	The combination of linezolid and fosfomycin was synergistic and bacteriostatic against four tested strains. Treatment of *Galleria mellonella* larvae infected with lethal doses of *S. aureus* resulted in significantly enhanced survival rates when low-dose of combination has no significant differences with high-dose combination. Combination therapy including linezolid and fosfomycin has synergistic effect against *S. aureus* in vitro and in an experimental *G. mellonella* model, and it suggests that a high dose of linezolid and fosfomycin may not be necessary.					
Chai et al., 2016	In vitro/in vivo preclinical study	3 MRSA strains	LIN 40 mg/kg q12 h + FOS 300 mg/kg q12 h	A FICI ≤ 0.5 was found for LIN + FOS combination, showing the best synergistic effect in all strains. The combination of LIN and FOS in a catheter-related biofilm rat model found that viable bacteria counts in biofilm were significantly reduced after treatment (*p* < 0.05).					
Xie et al., 2021	In vitro/in vivo preclinical study	One MRSA strain	LIN 2.5–10 mg/kg + FOS 50–200 mg/kg	Antibiotic combination showed excellent synergistic or additive effects on the original and the linezolid-resistant strain but showed indifferent effect for fosfomycin-resistant strain. In the *Galleria mellonella* infection model, the survival rate of the antibiotic combined was improved compared with that of the single drug. There was a good correlation between in vivo efficacy and in vitro susceptibility.					
**Necrotizing Fasciitis—Daptomycin ± Clindamycin**									
Samura et al., 2022	Systematic review with meta-analysis	7 studies (2 RCTs and 5 retrospective observational; 907 patients)	DAP vs. VAN	100% BSI due to MRSA (28–32.8% complicated SSTI)	DAP was associated with significantly lower mortality (OR 0.53, 95%CI 0.29–0.98) and higher treatment success (OR 2.20, 95%CI 1.63–2.96) compared to VAN. For intermediate-risk sources (including complicated SSTI), DAP was a factor increasing treatment success compared with VAN (OR 4.40, 95%CI 2.06–9.40).				
Cogo et al., 2015	Multicenter retrospective registry	1927	DAP ≥ 4–6 mg/kg/day	100% complicated SSTI	*S. aureus* 51.9% (MRSA 31.5%)	NA	Overall clinical success rate: 84.7% Clinical success rate in MRSA subgroup: 87.0%	NA	DAP treatment resulted in high clinical success rates in patients with different complicated SSTI subtypes, the majority of whom having failed previous antibiotic therapy, with no safety issues.
Gatti et al., 2019	Retrospective case-control study	62 (32 receiving IMM vs. 30 SM)	DAP 8–10 mg/kg/day vs. VAN 20 mg/kg/day	100% NSTI	*S. aureus* 22.6% (MRSA 3.2%)	ICU admission 100%MV 90.3% Vasopressors 83.9% Immunosuppression 24.2%	ICU mortality rate: 15.6% vs. 40% (*p* = 0.032) 7-day mortality rate: 3.1% vs. 20% (*p* = 0.049)	NA	IMM was more effective than SM as it allowed the earlier control of infection and the faster reduction of multiple organ-dysfunction (ΔSOFA −5.2 ± 3.5 pts. versus −2.1 ± 3.0 pts.; *p* = 0.003).

AE: adverse event; AKI: acute kidney injury; AUC: are under concentration-time curve; BJI: bone and joint infection; BSI: bloodstream infection; CAP: community-acquired pneumonia; CCI: Charlson comorbidity index; CEF: cefazolin; CI: continuous infusion; CNS: central nervous system; CR-BSI: catheter-related bloodstream infection; CSF: cerebrospinal fluid; DAP: daptomycin; EOT: end of treatment; FICI: fractional inhibitory concentration index; FOS: fosfomycin; GEN: gentamycin; HAP: hospital-acquired pneumonia; HR: hazard ratio; IAI: intrabdominal infection; ICU: intensive care unit; IE: infective endocarditis; IHD: intermittent hemodialysis; II: intermittent infusion; IMM: intensive multidisciplinary management; LIN: linezolid; LRTI: lower respiratory tract infection; MIC: minimum inhibitory concentration; MRSA: methicillin-resistant *S. aureus*; MRSE: methicillin-resistant *S. epidermidis*; MSSA: methicillin-susceptible *S. aureus*; MSSE: methicillin-susceptible *S. epidermidis*; MV: mechanical ventilation; NA: not assessed; NS: not significant; NSTI: necrotizing soft tissue infections; OR: odds ratio; PK: pharmacokinetic; PSI: pneumonia severity index; PVE: prosthetic valve endocarditis; RCT: randomized controlled trial; RIF: rifampicin; RR: risk ratio; SIRS: systemic inflammatory response syndrome; SM: standard management; SoC: standard of care; SSTI: skin and soft tissue infection; TDM: therapeutic drug monitoring; TEI: teicoplanin; UTI: urinary tract infection; VAN: vancomycin; VPA: ventilator-associated pneumonia; VISA: vancomycin-intermediate *S. aureus*.

## Data Availability

The data presented in this review are retrieved and summarized from the different publicly available included studies.

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
