# Peer review of "Targeted Therapy of Severe Infections Caused by Staphylococcus aureus in Critically Ill Adult Patients: A Multidisciplinary Proposal of Therapeutic Algorithms Based on Real-World Evidence"

_microorganisms, 2023, doi:10.3390/microorganisms11020394_

Round 1
Reviewer 1 Report
The maunscript presents a sound series of findings based on a literature search. I believe the scope nature of the literature search could have been better - details of search words could have been described and wider (line 90-103 was not exact in the "terms" used and the results...the number of articles found etc). Actually, with some of literature search words (line 90-103) that seemed to be used, there very clearly would have been more research articles found than were described in this manuscript. Indeed when mentioning the results from the literature search (line 151, line 163, for example) there is only one or two research examples provided (reference 18, or reference 24 and 25). This is discussing a very small section of a large research field. There are parameters for systematic reviews and while this manuscript is not proposing that this work is a systematic review, it falls short in presenting the research in the field through their literature search. They seem to have "picked" a few articles to focus on. This is missing other work on other antibiotics (vancomycin...) and treatment regimes and the development in-host antibiotic resistance. Also my own literature search quickly reveals that there is work in this area that highlights persistent infections (bactiremia, osteomyeltisis and others) with antibiotic tolerance and biofilm formation. I appreciate this area is likely outside the scope of the current work, but certainly in the background or setting up the project, these aspects should have been factored in. Also the authors mention an "algorithm" they are establishing - but this is not detailed in the end.
I think there is a good basis for the work in the manuscript, but some details could be elaborated.
Author Response
We thank the reviewer for this comment, allowing us to better clarify the purpose of our manuscript and the methodology that was applied. We fully agree with the reviewer that when performing systematic reviews specific criteria should be adopted, included the numbers of article found with the search strategy. However, systematic review was not the scope of our study. Our study was conceived for developing treatment algorithms for targeted therapy of S. aureus infections in critically ill patients based on consensus agreement within a multidisciplinary team. The different choices provided in these algorithms were supported by scientific evidence in terms of either efficacy or PK/PD features of the different agents. To make this approach more understandable, we provided more details in the Methods section by adding all of the different terms that were used in the search strategy (refer to Line 94-106). Although it could have seemed to the reviewer that the refs were “picked up”, indeed we may confirm that all of the real-world studies supporting the specific therapeutic choices were retrieved in the literature and were mentioned in the study according to a hierarchical scale of the study design, as reported in the evidence pyramid. Specific details of these studies are summarized in Table 1 and 2, and the most representative ones were analyzed in depth in the main text of the Results section.
We agree with the reviewer also on the fact that antibiotic tolerance and biofilm formation may be important features of staphylococcal infections. Consequently, we added a specific statement on this general issue in the Introduction section (refer to Line 59-64). However, these features usually concern chronic infections, namely prosthetic bone and joint infections, which were not included as infection type in our algorithms as they do not represent a frequent occurrence in the critically ill scenario. In our study, we took into account only acute staphylococcal infections, which are the main topics in the critically ill setting.
Finally, in regard to the therapeutic choices depicted in the algorithms (Figure 1 and Figure 2), they were described in each section of the Results with a summary of the supporting rationales based on real-world evidence and PK/PD features.
Reviewer 2 Report
General comments: The authors have developed two different algorithms with respect to targeted antimicrobial treatment for severe infections caused by Staphylococcus aureus and methicillin-resistant Staphylococcus aureus (MRSA). They categorized the therapeutic options for the respective site of infection and selected them based on the pharmacokinetic/pharmacodynamic features. Based on the algorithms Cefazolin or oxacillin was proposed for all of the different types of severe MSSA infections. The proposed targeted therapies for MRSA infections differed based on the infection site.
The authors present a very interesting study. The novel approach of establishing algorithms for the targeted therapies based on scientific evidence and optimization of the pharmacokinetic/pharmacodynamic will be a valuable strategy for treating MSSA as well as MRSA infections. The study has a sound methodology.The analysis has been described well. The discussion is clear and balanced.I would like to congratulate the authors for a job well done on the novel approach which will aid in the treatment of MRSA infections. I recommend the manuscript be accepted in its present form.
In addition, 'Staphylococcus aureus' has to be italicized throughout the manuscript.
Author Response
We thank the reviewer for this comment and for appreciating our work. As suggested, we italicized the term “Staphylococcus aureus” throughout the manuscript.
Round 2
Reviewer 1 Report
The changes are good and improve the manuscript